# Automatic Closed Captioning for Estonian Live Broadcasts

**Tanel Alumäe**
Tallinn University of Technology
`tanel.alumae@taltech.ee`

**Joonas Kalda**
Tallinn University of Technology
`joonas.kalda@taltech.ee`

**Külliki Bode**
Estonian Association of the Hard of Hearing
`kylliki.bode@gmail.com`

**Martin Kaitsa**
Estonian Public Broadcasting
`martin.kaitsa@err.ee`

## Abstract

This paper describes a speech recognition based closed captioning system for Estonian language, primarily intended for the hard-of-hearing community. The system automatically identifies Estonian speech segments, converts speech to text using Kaldi-based TDNN-F models, and applies punctuation insertion and inverse text normalization. The word error rate of the system is 8.5% for television news programs and 13.4% for talk shows. The system is used by the Estonian Public Television for captioning live native language broadcasts and by the Estonian Parliament for captioning its live video feeds. Qualitative evaluation with the target audience showed that while the existence of closed captioning is crucial, the most important aspects that need to be improved are the ASR quality and better synchronization of the captions with the audio.

## 1 Introduction

Deaf and hard of hearing (DHH) individuals face significant barriers when it comes to accessing live television broadcasts. Without closed captioning, they are unable to fully understand and engage with the content being presented. An automatic closed captioning system for live TV broadcasts would help to address this issue and provide DHH individuals with greater access to the same information and entertainment as their hearing counterparts. Closed captioning is not only beneficial for DHH individuals, but also for those who may have difficulty hearing the audio on their television due to background noise or other factors.

Until the beginning of 2022, Estonian Public Television (ETV) provided DHH-focused subtitles for some pre-recorded native language programmes, but not for live programmes. From

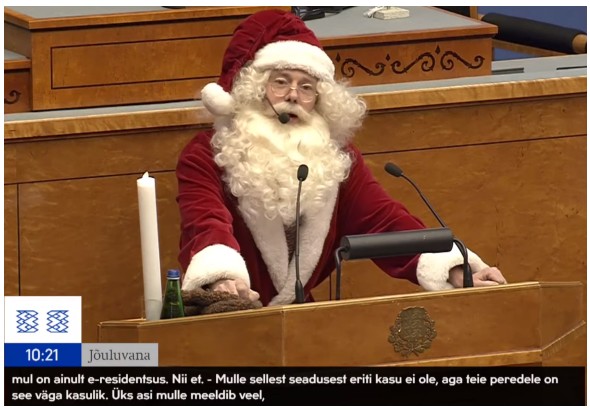

Figure 1: Closed-captioned live YouTube stream of the Estonian parliament.

March 2022, captions generated using automatic speech recognition (ASR) technology were added to the majority of live native-language programmes, such as news and talk shows. The same technology is used to provide closed captions to the live streams of the Estonian parliament sessions (see Figure 1). This paper describes the system used to generate the subtitles. We provide information on the architecture of the system, its different components, their training data and performance. We also summarise the results of a qualitative evaluation of the live captioning system carried out with the target audience, and discuss how the system could be improved.

The reported system is free and available under open-source license [1].

## 2 Previous Work

Real-time captioning systems based on speech recognition have been in use for several decades. Initially, such systems relied on so-called re-speakers - trained professionals who repeat what they hear in the live broadcast in a clear and ar-

---

[1] `https://github.com/alumae/kiirkirjutaja`

ticulate manner (Evans, 2003; Imai et al., 2010; Pražák et al., 2012). This allows supervised speaker adaptation of ASR acoustic models to be used, resulting in very accurate output. In some use cases, re-speakers also simplify and rephrase the original speech, instantly check and correct the resulting captions, and insert punctuation symbols. In some captioning systems, ASR is applied directly to the speech in the live programme, but a human editor is used to correct the ASR errors (Levin et al., 2014). However, training re-speakers and real-time editors is a long and expensive process. In addition, several re-speakers and/or editors are usually required, as one person cannot usually work continuously for more than two hours without a break.

As the quality of ASR systems has improved rapidly in recent years, there are more and more cases where an ASR system is used directly to produce subtitles without any post-processing. For example, ASR-based captions in multiple languages are available in online meeting platforms such as Zoom, Skype and Teams. Moreover, YouTube offers captioning for live streams, albeit exclusively in English at the time of writing. Streaming ASR for Estonian is available through several commercial vendors; however, recent evaluations have demonstrated that the ASR quality provided by these services falls short compared to the models developed at Tallinn University of Technology (Vapper, 2023).

## 3 Closed Captioning System

### 3.1 Architecture

Our closed captioning system consists of the following components:

1. Speech input: speech is either read from standard input (as a 16-bit 16 kHz PCM audio stream) or from a URL. Any stream type supported by *ffmpeg* is allowed, including video streams.

2. Audio stream is segmented into 0.25 second chunks and processed by a voice activity detection model which detects speech start and endpoints in the stream. We use the open source Silero VAD model (Silero Team, 2021), available under the MIT License;

3. Speaker change detection model indicates likely speaker change points in speech segments (see Section 3.2);

4. Each speaker turn is processed by a language identification module that filters out segments that are likely not in Estonian (Section 3.3);

5. Speech recognition, resulting in a stream of words tokens (Section 3.4);

6. Inverse text normalization (mostly converting text to numbers), implemented using handwritten finite state transducer rules using the Pynini library (Gorman, 2016);

7. Insertion of punctuation symbols and subsequent upperacasing (Section 3.5);

8. Confidence filter that hides decoded words that are likely to be incorrect (Section 3.6);

9. Presentation: displaying the captions or sending them to the API endpoint selected by the user (Section 3.7).

### 3.2 Speaker Change Detection

In order to make captions for dialogue more legible speaker change points need to be marked by a symbol such as "-". To detect change points we use an online speaker change detection model[2] which treats this as a sequence classification problem and labels each frame with either 1 or 0 depending on whether a speaker change happened or not.

The model is trained on an Estonian broadcast dataset detailed further in Section 3.4.1. Training is done on samples from speech segments with random lengths between 10 and 30 seconds. Background noise and reverberation are added to each segment both with a probability of 0.3. Background noises come from the MUSAN corpus (Synder et al., 2015). For reverberation, we used small and medium room impulse responses as well as real room impulse responses (Ko et al., 2017; Szöke et al., 2019). A classification threshold is learned on a 1-hour development split.

The model uses 1280-dimensional features obtained from a Resnet-based extractor (Alumäe, 2020) which is pre-trained on VoxCeleb2 (Chung et al., 2018). This is followed by two long short-term memory (LSTM) layers both with 256-dimensional hidden layers. A 1-second label delay is used since the model needs to see past the current frame to predict a change point. We use

---

[2] https://github.com/alumae/online_ speaker_change_detector

a collar-based loss function that encourages the model to predict a single positive frame in a 250ms neighborhood of an annotated change point. This training method has been shown to outperform the standard binary cross-entropy loss for the SCD task (Kalda and Alumäe, 2022). A further benefit of this loss function is that the model outputs develop peaks concentrated in a single frame. This removes the need for post-processing to find the exact timestamps of change points and decreases overall latency.

### 3.3 Language Identification

Broadcast news programs often contain foreign language segments, such as studio or field interviews. For those segments, no captions should be shown, since an Estonian ASR system doesn't produce meaningful output for speech in other languages. Furthermore, foreign language video segments in television news programs often already have Estonian subtitles and automatic captions would interfere with them.

For filtering out non-Estonian speech segments, we first process the first three seconds of every speech turn using the open source Silero language identificaton model (Silero Team, 2021), available under the MIT License. During the initial development phase, we found that the first 3 seconds are sometimes unreliable for language detection, since they often contain hesitation and/or other paralingusitic speech sounds that confuse the language detection model. Therefore, if a turn is rejected based on the first three seconds, another test is performed using the first five seconds of the turn. If this test also indicates that the speech is not in Estonian, the whole speech turn is ignored by the rest of the pipeline and no captions are produced for this speaker turn. Of course, this assumes that a speaker doesn't change the language during a single turn which might not always be true.

The language classifier that we use discriminates between 95 languages and claims 85% validation accuracy. However, we are not interested in the actual language spoken in the segments, but only in the the fact whether the segment is in Estonian or not. This allows us to use a simple method to increase the robustness of the language classifier. Namely, we assume that our system is always used on streams where the input language is mostly in Estonian, which means that the prior probability of Estonian is much higher than the de-

| Source | Amount (h) |
|---|---|
| Broadcast speech | 591 |
| Spontaneous speech (Lippus, 2011) | 53 |
| Elderly speech corpus (Meister and Meister, 2022) | 49 |
| Talks, lectures | 38 |
| Parliament speeches | 31 |
| Total | 761 |

Table 1: Acoustic model training data.

| Source | Tokens (M) |
|---|---|
| ENC19 Web Scrape | 526 |
| ENC19 Ref. Corpus | 185 |
| ENC19 Wikipedia | 35 |
| OpenSubtitles | 98 |
| Speech transcripts | 6.1 |
| Subtitles from ETV | 3.8 |
| Total | 854 |

Table 2: Language model training data.

fault uniform probability $P_u(l) = 1/95$. Therefore, we "fix" the conditional probability distribution $P(l|x)$ returned by the language identification model for input segment $x$ to use the appropriate prior:

$$P'(l|x) = \frac{\frac{P'(l)}{P_u(l)} \times P(l|x)}{Z}$$

where $Z$ is a normalizing factor and $P'(l)$ is the prior probability for languages:

$$P'(l) = \begin{cases} P'(l = \text{et}), & \text{if } l = \text{Estonian} \\ (1 - P'(l = \text{et}))/94, & \text{otherwise} \end{cases}$$

Based on small-scale finetuning, we use a prior probability $P'(l = \text{et}) = 0.5$ for Estonian.

### 3.4 Speech Recognition

#### 3.4.1 Data

Speech data that is used for training the speech recognition acoustic model is summarized in Table 1. Only the duration of the segments containing transcribed speech is shown, i.e., segments containing music, long periods of silence and untranscribed data are excluded.

Most of the training data has been transcribed by our lab in the last 15 years (Meister et al., 2012), except the Corpus of Estonian Phonetic Corpus of Spontaneous Speech that originates from the University of Tartu (Lippus, 2011).

Textual data used for training the language model (LM) is listed in Table 2. Most of the data originates from the subcorpora of the Estonian National Corpus 2019 (ENC2019) (Kallas and Koppel, 2019): Estonian web, a reference corpus containing balanced data from the web, newspapers and books, and Estonian Wikipedia. We also use all available Estonian data from the OpenSubtitles corpus (Lison and Tiedemann, 2016) and scraped DHH subtitles from ETV.

Before using the text data for LM training, text normalization is performed. Texts are tokenized, split into sentences and recapitalized, i.e., converted to a form where names and abbreviations are correctly capitalized while normal words at the beginning of sentences are written in lower case. This is done with the help of the EstNLTK morphological analyzer (Laur et al., 2020). Numbers and other non-standard words are expanded into words using hand-written rules.

### 3.4.2 Models

The ASR model is implemented using Kaldi (Povey et al., 2011). The acoustic model is a factored time-delay neural network (TDNN-F) acoustic model (Povey et al., 2018) with six convolutional layers and 11 TDNN-F layers. The acoustic model has around 17 million parameters. Online speaker adaptation is done using i-vectors. We use standard Kaldi multi-condition data augmentation (Ko et al., 2017) for acoustic training data: training data is 3-fold speed perturbed, and the speed perturbed data is in turn augmented with reverberation, various environment sounds, music or babble noise from the MUSAN corpus (Synder et al., 2015). This increases the amount of training data by 15-fold in total. The acoustic model is trained for four epochs on the augmented data.

The LM of the system uses $200\,000$ compound-split units (i.e., compound words are broken to constituents). It is an interpolation of 4-gram sub-models trained on each of the subcorpora, with interpolation coefficients optimized on development data. The final model is pruned so that the resulting HCLG transducer would allow decoding with 16 GB of RAM. After decoding, we apply out-of-vocabulary (OOV) word recovery to reconstruct the orthographic transcripts of the decoded unknown words. Compound words are reconstructed from the decoded constituents using a hidden-even n-gram model (Alumäe, 2007). Various specifics of language modeling are described in more de-

|  | WER |
| --- | --- |
| TV news | 8.5 |
| Talkshows | 13.4 |
| Press conferences | 8.1 |

Table 3: Word error rate of the ASR system on various speech data.

tails in (Alumäe et al., 2018).

We validated the performance of the models on a dedicated test set collected especially for this project. It consists of TV main evening news, casual TV talkshows, and press conferences of the Tallinn city council and the state's health board, with a total duration of 12 hours. Table 3.4.2 shows the word error rate (WER) of the ASR system on each subcorpus. As can be seen, TV news and press conferences produce noticably less ASR errors than talkshows, which is probably related to the higher degree of spontaneousness in talkshow speech.

The decoding module is implemented using a forked version of the Vosk Speech Recognition Toolkit[3] that supports word timestamps for intermediate recognition hypotheses.

Closed captions on television generally do not offer verbatim speech transcriptions, particularly for spontaneous speech. Elements such as repetitions, hesitations, pause fillers, false starts, and interjections are typically omitted from the captions, and sentences are reformatted to ensure grammatical correctness. Presently, our system lacks any modules to implement such modifications on the generated ASR transcripts. Only filled pauses and hesitations are excluded from the captions, since they are not transcribed in the ASR training data.

### 3.5 Punctuation Insertion

In order to make the captions more readable, the decoded stream of words is enriched with punctuation symbols. This is done using an LSTM model[4]. The model is trained on a mixture of speech transcripts from our ASR training corpus and a random sample of the LM training data, totalling in around 50 million words. The punctuation model operates on BPE-tokenized text, using a BPE vocabulary of 100K tokens. The model first projects the input tokens into 512-dimensional embeddings and then applies four unidirectional

---

[3] https://github.com/alphacep/vosk-api
[4] https://github.com/alumae/streaming-punctuator

LSTM layers, with a hidden layer dimensionality of 512. For token corresponding to word endings, the most likely punctuation symbol is predicted from the vocabulary of [*None*, ".", ",", "?", "!"]. A label delay of two is used, i.e., at each time step, the model predicts a punctuation symbol for a token two timesteps in the past. This effectively allows the model to predict a punctuation symbol, given the past tokens and two upcoming tokens.

The model was validated on the transcripts of the ASR validation set and resulted in a F1 score of 72%, micro-averaged across all punctuation marks.

### 3.6 Confidence Filter

In some situations, such as severe background noise, overlapped speech or very spontaneous speech, the quality of the ASR output degrades significantly. In such cases, it is preferable not to show any captions at all, since they are practically useless for understanding the content of the speech and bring a lot of confusion to the viewer. Therefore, the closed captioning system includes an additional component that tries to hide captions segments that are likely wrong.

The ASR decoder that we use outputs word confidence values for all decoded tokens. The confidence scores are computed by the Kaldi decoder from the confusion network of the Minimum Bayes Risk (MBR) decoding result (Xu et al., 2011). Since such confusion scores are often not very reliable, the captioning system observes the averaged confidence scores of the words calculated over a five word window, and hides words whose averaged confidence score falls below a threshold (we use a threshold of 0.75). Evaluating, finetuning and calibration of this component remains currently for future work.

### 3.7 Presentation

The system can present the generated captions in a variety of formats and modes. Currently, it supports several commercial captioning delivery platforms as well as YouTube live streaming. Most media streaming platforms that support closed captioning expect word-by-word captions: i.e., captions should be provided on a word-by-word basis (possibly with a timestamp), and words already displayed cannot be changed. This poses some challenges for our captioning system, as several factors cause the final part of the caption to change dynamically: new words coming from

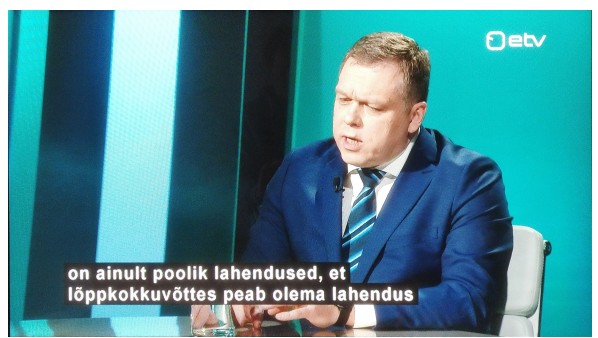

Figure 2: Closed-captioned ETV talk show.

the decoder may cause already decoded words to change (e.g. due to word to number conversion), punctuation may be inserted before the already decoded word (due to the two-word label delay of the punctuation model). For this reason, the caption presentation module includes functionality to delay the final output of generated words to the currently used subtitle transmission platform until it is certain that the word won't change. This (and the delay caused by the speaker change detection model and the language identification model) results in a delay of approximately 3-5 seconds relative to the speaking time of the words, which can be mitigated by also delaying the transmission of the multimedia stream. For those presentation modes that allow dynamically changing captions, a much lower delay or approximately 2 seconds is possible.

## 4 Integrations

At the time of writing this paper, the closed captioning system is used by the Estonian Public Television (ETV) and by the Estonian parliament.

In ETV, the captioning system runs continuously, but the captions are actually delivered only to specific native-language programs (see Figure 2). The system outputs captions on a word-by-word basis to special caption transmission software that formats the words into caption lines and blocks. Due to the approximately 5-second delay in the video signal caused by the encoding process, the captions and video are roughly synchronized, but the synchronization is currently not exact: captions tend to be delayed at the beginning of a speaker's turn and arrive relatively faster at the end of a turn.

Closed captions are transmitted on a dedicated DHH digital closed captioning channel and are not displayed by default. End users can enable closed

captioning from the user interface of their device.

## 5 Qualitative Evaluation

### 5.1 Introduction

In order to better understand, how the automatically generated captions on ETV are used and experienced, and what are the most outstanding shortcomings, we conducted a qualitative evaluation with the intended focus group of the technology. The purpose of this study was to investigate the following research questions:

- How often do DHH individuals use the ASR-generated closed captions on ETV?

- How do the closed captions improve the quality of life of DHH individuals?

- Which aspects of the system need to be improved?

### 5.2 Methodology

This research study used semi-structured interviews to gather information. Since most of the study participants were DHH individuals, three of the interviews were conducted via e-mail, one via a chat application, and only one via telephone. Additional user feedback was collected via Facebook in response to a call for comments by a person followed by a large DHH community.

Participants were sought through personal contacts of the researcher. Four of the participants were hard-of-hearing individuals and one didn't have any significant hearing loss. Data collection took place in January 2023.

### 5.3 Findings

Three participants with hearing impairments acknowledged that their hearing loss prevents them from understanding speech on television, even when using a hearing aid. One individual with hearing impairment mentioned that she could comprehend the speech if the TV volume were significantly increased, but she refrains from doing so as she is the only person with hearing impairment in her family. All the DHH participants reported that they activate automatic subtitles whenever they watch live ETV broadcasts daily. The only participant without hearing loss revealed that he watches programs with automatic subtitles multiple times a week when ambient household noise or background conversations make it challenging to hear the television audio.

All the participants highlighted the importance of having subtitles. Several people reported that it enabled them to watch television with their family. One participant expressed that subtitles helped her feel included in society and enabled her to stay better informed about events occurring in the country.

All participants stressed that the most crucial aspect of the existing captioning system requiring improvement is its accuracy. One participant highlighted that the quality of captions is currently good for TV presenters, but often falls short for "ordinary people" (i.e. interviewees on news and talk shows). One person explained that although it is sometimes difficult to understand what is actually being said (due to ASR errors), it is still important to have the captions. The second aspect that was often highlighted was that in the current captioning system, the captions are often not well synchronized with the audio (as opposed to manually created subtitles). Other issues raised include misrecognition of named entities, poor marking of speaker turns, occasional dropping of the captions (i.e., when the confidence filter is activated) and the fact that subtitles sometimes interfere with other information on the screen, such as speaker names. Several respondents also noted that subtitles are not currently available for all native language broadcasts.

## 6 Conclusion and Future Work

This paper described an ASR-based realtime closed captioning system for Estonian broadcasts. The system consists of several open-source components and is currently used for providing captions to Estonian public television native language broadcasts and for captioning the live streams of the Estonian parliament.

Qualitative evaluation with the hard-of-hearing focus group showed that providing captions to live TV broadcasts is of high importance to this community. The study emphasized that it is urgent to further improve the ASR quality of the closed captions and to improve the synchronization between audio and captions.

We are currently working on several aspects of the system that would address some problems highlighted in the qualitative evaluation. First, we are preparing to migrate to end-to-end streaming transducer ASR models that would provide improved accuracy with relatively low latency. We

are also experimenting with integrating the decoding of punctuation symbols to the main ASR model, since currently the separate punctuation symbol insertion model is a source of around two second latency in subtitle presentation. Also, we have already implemented modifications to the system that would allow exact synchronization between the audio and displayed captions.

## Acknowledgments

This research has been supported by the Centre of Excellence in Estonian Studies (CEES, European Regional Development Fund). The authors acknowledge the TalTech supercomputing resources made available for conducting the research reported in this paper.

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
