# OpenReview forum: "Automatic Closed Captioning for Estonian Live Broadcasts"
_NoDaLiDa/2023/Conference — NoDaLiDa 2023_

### Official Review · Reviewer_9oKr · 2023-03-06
**Review of paper 87**

**Rating:** 8
**Confidence:** 5

**Review:**

This paper describes a speech recognition based closed captioning system for Estonian language, primarily intended for the hard-of-hearing community. Authors describe all core components of the system and  also summarise the results of a qualitative evaluation of the live captioning system carried out with the target audience.

The paper is well written and easy to read. The method as applied in the paper is described well. The architecture, optimization, data and technical challenges are clearly described.

Strengths&weaknesses:
- [+] All components of the system are described in details: architecture, data, design choices, some evaluation.
- [+] The idea of detecting change speaker instead of full diarization is interesting.
- [+] Qualitative evaluation of real user experience.
- [+] Great effort on making all components of the system public and open-source.
- [-] Solution mostly consists of reusing existing architectures and open-source components. Lack of novelty.
- [-] Lack of comparison with other system or components. For example, how does new punctuator compare with previous work?
- [-] It's unclear how many respondents participated Qualitative Evaluation. It seems there were only 5 of them.
- [-] It would be good to see how much latency each component adds to the system (i.e. some latency breakdown table).

Some comments:
- Given the fact that there are no rescoring and special adaptations are made in all components of the system, in my opinion, the latency of 5 seconds seems to be high. What is the main the reason for this? How does this compare with related work?
- Line 138 contains placeholder "FIXME: citation needed". Please replace with actual citation.

**Paper Type:**

Long paper

---

### Official Review · Reviewer_HRTq · 2023-03-09
**Automatic Closed Captioning for Estonian Live Broardcasts**

**Rating:** 9
**Confidence:** 4

**Review:**

The paper discusses a real-time closed captioning system for Estonian broadcasts that uses ASR technology. The system employs multiple open-source components and is presently utilized to deliver captions for broadcasts on Estonian public television, as well as to caption live streams from the Estonian parliament.
This is a very good paper that describes the real-time closed captioning system. It is well-written, clear, and definitely helpful for anybody interested in real-time closed captioning. The paper presents all system components, methods and data used and gives evaluation results.

Still, I have a few suggestions:

1. Authors claim that their results are available under an open-source license, which is not exactly true. There is no license provided in the referenced GitHub repositories. No license does not mean an open-source license, software without the license is not open-source software. I would recommend adding the license.

2. It is obvious that it is almost impossible to fully anonymize such a paper. It refers to GitHub repositories and domain experts can guess the author anyway. It is fine for me. I think the authors have done almost all that is necessary. Still, I would recommend avoiding using phrases that explicitly reveal you. For example -  "our laboratory" in line 320.

3. There are a few spelling errors, see lines 391 (TV --> of TV) , 448 (that a --> that are), 534 (indended).


**Paper Type:**

Long paper

---

### Official Review · Reviewer_ZtDC · 2023-03-09
**Good overview paper of a publicly deployed ASR system**

**Rating:** 8
**Confidence:** 4

**Review:**

The paper provides an overview of an ASR system used for captioning Estonian live broadcasts, outlining the main components (language identification, punctuation insertion, formatting of numerical expressions, etc.). The main challenges (latency, dealing with low confidence recognition results) are also presented. The system is in actual use by the Estonian TV allowing the authors also to summarize the user feedback.

The paper is easy to read and technically sound.

I would have liked to see a more detailed comparison against similar systems, both for Estonian and other languages. Also, the paper could describe more the linguistic aspects of the target language of the system, and how it differs from targets of other ASR systems, e.g. is there a need to transcribe every word (incl. interjections, pause fillers, etc.), does the TV channel follow certain standards for formatting date, time and other expressions that the system should follow or allow to be configured? The paper should also describe how to system is maintained, e.g. can the customer (TV station) fine-tune the system given a growing body of (manually corrected) transcriptions, or custom language model components (e.g. lists of named entities and other non-acoustic metadata).

**Paper Type:**

Long paper

---

### Decision · Program_Chairs · 2023-03-17

Accept